# A GaN HEMT Amplifier Design for Phased Array Radars and 5G New Radios

**DOI:** 10.3390/mi11040398

**Published:** 2020-04-10

**Authors:** Dawid Kuchta, Daniel Gryglewski, Wojciech Wojtasiak

**Affiliations:** Institute of Radioelectronics and Multimedia Technology, Warsaw University of Technology, Nowowiejska 15/19, 00-662 Warsaw, Poland; dgrygle@ire.pw.edu.pl (D.G.); wwojtas@ire.pw.edu.pl (W.W.)

**Keywords:** power amplifier, GaN 5G, high electron mobility transistors (HEMT), new radio, RF front-end, AESA radars, transmittance, distortions, optimization

## Abstract

Power amplifiers applied in modern active electronically scanned array (AESA) radars and 5G radios should have similar features, especially in terms of phase distortion, which dramatically affects the spectral regrowth and, moreover, they are difficult to be compensated by predistortion algorithms. This paper presents a GaN-based power amplifier design with a reduced level of transmittance distortions, varying in time, without significantly worsening other key features such as output power, efficiency and gain. The test amplifier with GaN-on-Si high electron mobility transistors (HEMT) NPT2018 from MACOM provides more than 17 W of output power at the 62% PAE over a 1.0 GHz to 1.1 GHz frequency range. By applying a proposed design approach, it was possible to decrease phase changes on test pulses from 0.5° to 0.2° and amplitude variation from 0.8 dB to 0.2 dB during the pulse width of 40 µs and 40% duty cycle.

## 1. Introduction

Radar systems, mainly 3D Active Electronically Scanned Array (AESA), strongly supported by the latest achievements in information technology, bring new challenges to the designers of transmit/receive (T/R) modules, especially High-Power Amplifiers (HPAs) based on solid-state devices [1,2]. In addition, there are currently rapid advances in high-speed wireless technology, such as 5G [3,4,5,6,7,8,9]. In particular, the power amplifiers, as a key element of RF transmitters, directly and significantly affect the operation quality of modern wireless communication systems and new generation radars [2,3,9]. The requirements concerning linearity and efficiency of HPAs are confronted with the needs of both systems for higher output power and improved the efficiency of heat management. In case of AESA, due to very complex beamforming techniques used, the strong emphasis is put on amplitude and phase constancy of the amplifier’s transmittance during the RF pulse and pulse-to-pulse [10], as in pulse the signal is increasingly more often modulated not only in frequency but also in amplitude and phase [11]. The same is true for the 5G and Long-Term Evolution Advanced (LTE-A) network systems which are using quadrature amplitude modulation (QAM) of higher orders and orthogonal frequency division multiplexing (OFDM) methods [11,12,13]. Both QAM and OFDM are particularly sensitive to transmittance changes generated mainly by output stages of base station transmitter power amplifiers [8,9,10,11,12,13]. Currently, such amplifiers must be linearized to meet wireless transmission standards defined e.g., by following parameters: in-band error vector magnitude (EVM), adjacent channel power ratio (ACPR), or the shape of spectrum mask [14,15,16]. There are many techniques for the amplifier linearity improvement such as e.g., analog feedforward, digital pre-distortion [6], dynamic biasing [7], envelope tracking [10], or Chireix’s outphasing method [17]. However, only a few of them are suitable for linearization of amplifiers operating with wideband spectrally efficient signals and high peak-to-average power ratio (PAPR) like LTE or 5G [5,9,18]. The common linearization technique applied in broadband transmitters of contemporary wireless systems is the baseband digital pre-distortion technique (DPD) preceded by the shaping of waveform crest factor (CF) [6]. All these methods have restrictions on applicability and require additional external hardware and/or software implementation consuming system resources. To reduce the system resource consumption, the nonlinearities of amplifiers should be as small as possible. There is a new challenge facing amplifier’s designers in developing new amplifier solutions for modern applications, like AESA, LTE, 5G as well as next-generation systems [2,3,4]. Our research just goes in this direction to examine amplifier transmittance changes as a function of load and source impedance of a transistor. Since the transmittance changes are not only a symptom of transistor nonlinearity but they are also a response to the temperature changes inside a transistor due to the complexity of amplified signals. It is known that the thermal effects in the transistor active layer are essentially responsible for the amplitude distortion while variations of internal time delays inside the transistor are the main reason for a phase distortion during quick signal envelope changes in time, e.g., in pulse [19]. Therefore, it is necessary to use large-signal electro-thermal models [20]. The paper presents the dependencies of changes in the transmittance phase and magnitude on the GaN high electron mobility transistors (HEMT) load impedance. The design strategy based on the derived relationships and numerical simulations are confirmed by sophisticated time-domain measurements.

Our goal was to develop a temperature-dependent HPA modeling technique under operating conditions with signals of a variable envelope with built-in a function of transistor load impedance optimization for minimal transmittance distortions. For this purpose, the typical power amplifier structure was analyzed to find origins and relationships of changes in amplifier transmittance during radar pulse as well as in the defined time window in case of broadband wireless communication signals. The minimization of HPA transmittance changes will facilitate the use of correction methods in baseband, such as DPD and radar calibration.

In the paper, the proposed method is particularly focused on GaN HEMTs but it can be generalized to other types of transistors. There are few publications regarding GaN-on-Si HEMTs despite their price is twice as low as GaN-on-SiC. Thus, for the purpose of this work GaN-on-Si HEMT was chosen to examine its performance.

## 2. Power Amplifier Analysis

In order to identify sources of transmittance variations, the complete amplifier structure with the GaN HEMT was measured and analyzed [21,22]. On this basis, the relationship between the transistor load impedance and amplifier transmittance changes have been derived. We also want to show what the changes in the transmittance phase and magnitude depend on. The amplifier model schematic circuit to be analyzed is shown in Figure 1. For this purpose, it is sufficient that we use a small-signal model. The model represented by the equivalent circuit shown in Figure 1 was described by a set of parameters dependent on current and voltage values at the transistor operating points i.e., its parameters were extracted at the properly selected transistor operating conditions. 

It consists of a typical GaN HEMT transistor model with parameters determined at the average drain current *I_DA_* corresponding to a saturated output power. Other transistor model components invisible in Figure 1 are included in the source impedance *Z_s_* = *R_s_* + *jX_s_*, (*X_s_* > 0) and the load admittance *Y_L_* = *G_L_* + *jB_L_*. The source impedance is connected in series with the RF signal source modeled as an ideal voltage source *U_S_*. For clarity of our analysis, Miller theory is applied but there is a problem with determining an internal voltage gain *K_unit_* [23]. In general, the gain *K_unit_* should be calculated with account of all elements of the equivalent circuit shown in Figure 1. However, for low frequencies and roughly estimation for higher frequencies the internal voltage gain *K_unit_* can be simplified in the first approximation to the following form:(1)$$K_{uint}=\frac{U_{L}}{U_{gsi}}=-\frac{g_{m}}{G_{L}^{\prime }}$$
under the following assumption:(2)$$R_{S}+R_{in}>\frac{1}{\omega \left[C_{gs}+\left(1+\frac{g_{m}}{G_{L}^{\prime }}\right)C_{gd}\right]}\;or\;X_{S}\ll \frac{1}{\omega \left[C_{gs}+\left(1+\frac{g_{m}}{G_{L}^{\prime }}\right)C_{gd}\right]},\;$$
where:(3)$$G_{L}^{\prime }=G_{L}+\frac{1}{R_{ds}}\;.$$

The simplified HEMT model schematic circuit takes the form as shown in Figure 2.

The Miller theorem expresses only equivalents of *C_gd_*; *C_gs_* and *C_ds_* are parallel to these equivalents, thus they can be added giving *C_in_* and *C_out_*. Hence the input and output equivalent Miller’s capacitances are given as:(4)$$C_{in}=C_{gs}+C_{dg}\cdot \left(1+\frac{g_{m}}{G_{L}^{\prime }}\right)$$
(5)$$C_{out}=C_{ds}+C_{dg}\cdot \left(1+\frac{G_{L}^{\prime }}{g_{m}}\right).$$

When the amplifier is operated close to saturated output power, the ratio *G’_L_*/*g_m_* is much smaller than unity as given in Table 1. Under this assumption the Equation (5) can be simplified to the following form:(6)$$C_{out}\approx \;C_{ds}+C_{dg}.$$

Parameters of three L-band GaN HEMTs, provided by the manufacturer are given in Table 1. They were calculated at the average drain current *I_DA_* for output power close to *P_sat_* and implemented in Keysight software ADS.

When the GaN HEMT is operated as the current source the output capacitance is almost constant depending on the drain-to-source voltage *U_DS_* and RF signal amplitudes in a wide range [8]. For the sake of clarity, it was assumed that all frequency harmonics are shorted as for ideal class A and AB [24]. To eliminate the imaginary part of output admittance in the transistor model the load susceptance *B_L_* should be as follows:(7)$$B_{L}=-\omega C_{out},$$
which leads to the amplifier model schematic diagram as shown in Figure 3.

Applying the Kirchhoff voltage law to the model from Figure 3, the transmittance was derived. The transmittance phase and magnitude are expressed by (8) and (9). These formulas reveal details of the transmittance dependence on transistor parameters and parameters of the matching networks.
(8)$$arg\left(\frac{U_{L}}{U_{S}}\right)=arg\left(j\frac{g_{m}}{G_{L}^{\prime }\omega C_{in}}\cdot \frac{1}{R_{S}+R_{in}+j\left(X_{S}-\frac{1}{\omega C_{in}}\right)}\right)$$
(9)$$\left|\frac{V_{L}}{V_{S}}\right|=\frac{g_{m}}{G_{L}^{\prime }\omega C_{in}}\cdot \frac{1}{\sqrt{{\left(R_{S}+R_{in}\right)}^{2}+{\left(X_{S}-\frac{1}{\omega C_{in}}\right)}^{2}}}.$$

Equations (8) and (9) show that changes in the amplifier transmittance magnitude are influenced by both transistor parameters as well as the structure and parameters of matching networks.

Obviously, the transistor parameters in Equations (8) and (9) depend on temperature [19,25,26]. This explains the impact of temperature on the transmittance changes during pulse transfer as well as under the large amplitude variations of the signal with higher PAPR. In both cases, due to the high signal amplitude, the dynamic power dissipated in the transistor also strongly varies in time. As a response the temperature inside the active layer of the transistor is changed which proves the first thesis of this work.

Moreover, Equations (8) and (9) show that the transmittance phase increases monotonically with the change of load conductance *G_L_*. Test results obtained in [27] confirm a validity of the relations (8), (9). To increase the accuracy of transmittance model the output capacitance *C_out_* was taken into account. The phase and magnitude are given by (10) and (11) accordingly.
(10)$$arg\left(\frac{V_{L}}{V_{S}}\right)=arg\left(j\frac{g_{m}}{\left(G_{L}^{\prime }+j\left(B_{L}+\omega C_{out}\right)\right)\omega C_{in}}\cdot \frac{1}{R_{S}+R_{in}+j\left(X_{S}-\frac{1}{\omega C_{in}}\right)}\right)$$
(11)$$\left|\frac{V_{L}}{V_{S}}\right|=\frac{g_{m}}{\omega C_{in}}\cdot {\left(\begin{array}{c} {\left(\left(R_{S}+R_{in}\right)\cdot G_{L}^{\prime }-\left(X_{S}-\frac{1}{\omega C_{in}}\right)\cdot \left(B_{L}+\omega C_{out}\right)\right)}^{2}\\ +{\left(\left(R_{S}+R_{in}\right)\cdot \left(B_{L}+\omega C_{out}\right)+\left(X_{S}-\frac{1}{\omega C_{in}}\right)\cdot G_{L}^{\prime }\right)}^{2} \end{array}\right)}^{-\frac{1}{2}}.$$

The Equations (8)–(11) clearly show the dependence of the transmittance on the load impedance *Z_L_*, which can be determined during the amplifier design.

In conclusion, using Equations (8) and (9) it is possible to facilitate initial steps of power amplifier design for the minimal transmittance changes. Using (8)–(11) formulas we can estimate the level of reduction of the transmittance changes by tuning *Z_S_* and *Z_L_* impedances. In our case, these values are 0.254° and 0.4 dB, for phase and magnitude, respectively, and are consistent with the simulation results which are 0.3° and 0.6 dB. Although this is a qualitative analysis, it is quite well in agreement with quantitative simulations using advanced software.

## 3. Measurement Setup and Amplifier Modelling

As a part of the research, measurements of waveforms of power amplifiers were performed. These measurements were performed using the Keysight DSAV334A Infiniium V-Series digital signal analyzer (DSA) (Keysight, Santa Rosa, CA, USA) and Keysight N5172B EXG X-Series vector signal generator with the option of generating training pulses. The purpose of such measurements was to examine the amplitude and phase distortions caused by the amplifier. The test program assumed the use of four pulse trains with a carrier frequency f_0_, which are illustrated in Figure 4.

The measurement consisted of recording in the time domain, previously designed, four train pulses, using DSA. To detect changes in the signal handled by the amplifier under test, the training signal was recorded before and after passing through the amplifier, working with a power close to *P_1dB_* region. For this reason, the signal from the generator was split into two paths using HP11667A power divider (HP, Palo Alto, CA, USA). One of the paths was directly connected to DSA while the second one was connected to the amplifier input. The output of the amplifier was connected to the second DSA channel via the HP778D directional coupler. To protect the measurement instruments in both paths proper attenuators were used (the impact of attenuation on the obtained results was checked). The measurement setup is shown in Figure 5.

The sample measurement results are shown in Figure 6. In the upper part of Figure 6 the input waveform is presented, in the middle, the output waveform coming directly from the amplifier under test and in the lower part the difference between these waveforms is calculated as a change of the transmittance phase during the pulse duration.

Based on the measurements, derived formulas, and using the Envelope simulation technique available in Keysight ADS software, we developed the power amplifier design method for minimization of the transmittance changes. To develop an algorithm for determining optimum load impedance, a test amplifier with 14W GaN HEMT NTP2018 from MACOM (Lowell, MA, USA) was designed using the large-signal transistor model provided by MACOM. The test amplifier was designed according to *Cripps* design methodology to achieve a trade-off between maximum output power and high efficiency [24]. The MACOM model is closed, and enables only the temperature characteristics under thermal steady-state conditions to be calculated. Therefore, for the purpose of this work, we developed our own large-signal model based on the Angelov [28]. Our model includes an extensive thermal part representing by 5-6 serially connected (*R_thi_*, *C_thi_*) parallel cells. The values of Angelov model parameters were fitted to obtain simulations consistent with the original non-linear MACOM model. The model thermal parameters were determined by fitting RC-ladder network to the thermal impedance *Z_th_(t)* of the transistor with was measured by the modified *DeltaU_gs_* method [29,30]. To determine load impedances the load-pull method was applied in ADS Software. The high compliance of the simulations and measurements of the amplifier was achieved.

An appropriate use of the Envelope simulation [31] together with the large-signal electro-thermal model allows for simulating and modeling of amplifier’s transmittance changes. The simulation provides great opportunities to optimize the structure of the amplifiers and facilitates the process of finding the optimal load impedance by viewing time waveforms. There are no studies in the literature that combine the electro-thermal model with the envelope simulation to study changes in the parameters of the transmitted signal. However, so far, a load impedance optimization of microwave amplifiers has been performed only under quasi-static conditions using simulations in the frequency domain. This made it impossible to analyze changes, in the signal during operation caused by widely understood memory effects at the stage of amplifier design. Moreover, optimization of amplifiers’ parameters (e.g., ACPR) generally was based on the appropriate selection of load impedance, during load-pull measurements, in order to determine its value for which the optimized parameters reach a satisfactory range [15,32,33,34]. This approach is unenforceable in case of transmittance changes that are dynamic over time due to the phase measurement, requiring an accurate calibration of the reference track with a phase shifter that has to be perfectly synchronized with the load-pull tuners. 

## 4. Test Amplifier

To show the proposed design methodology, step by step, the test amplifier with GaN HEMT NTP2018 was designed. The amplifier achieves more than 17 W of output power over a 1 GHz to 1.1 GHz frequency range, while MACOM in datasheet provides the information about 14 W. Assembly drawing and photo of the fabricated amplifier is shown in Figure 7 and Figure 8, respectively. The amplifier was fabricated on Rogers RO4003C laminate ($\varepsilon_{r}=3.55$, h = 0.020′’, T = 1oz.). 

To generate a sequence of training pulses of a given power, as a stimulus an internally matched simulation port was used. This enabled us to simulate the pulses transferred by the amplifier in the time domain. The average pulse power $P_{AVout}\left(t\right)$ over a time interval ∆t was calculated according to the following formula [35]:(12)$$P_{AVout}\left(t\right)=\frac{1}{∆t}\int_{0}^{∆t}PDF_{P_{0}}\left(t\right)\cdot P_{out}\left(t\right)dt,$$
where PDF denotes Probability density function (or the probability distribution function) of the output power. The simulation ADS schematic is shown in Figure 9.

The simulation was performed for two pulses with a duration of 40 μs and a period of 100 μs at the carrier frequency f = 1.05 GHz being the center frequency of the amplifier working band. In order to simulate the amplifier operating with output power close to P1dB the excitation power was set to 25 dBm in pulse. The test amplifier was characterized by the measurement setup shown in Figure 5 with the same parameters as given above. GaN HEMT NPT 2018 was biased at the quiescent operating point U_DS_ = 50 V and I_DQ_ = 75 mA. The simulation and measurement results of pulses at the amplifier output for frequency f = 1.05 GHz are shown in Figure 10.

The presented approach allows for simulating the amplitude and phase changes of the amplifier transmittance during pulse in a wide range of transistor load impedance. It is the basis to develop an algorithm for optimizing of the source and load impedances for minimal transmittance distortions.

To find the optimum impedance of *Z_LT_*, the Envelope simulation was used. It allows for modeling the amplifiers with very different waveforms and observing the waveforms at the amplifier output depending on the input power levels. With access to large-signal electro-thermal models and envelope simulation, it is possible to calculate the phase and amplitude changes caused by the amplifier during the pulse transfer. The range of phase and amplitude variations depends on the load impedance. For the optimum impedance *Z_LT_* this range is the smallest. The impedance was adjusted using load-pull tuners. The ADS schematic used for the load impedance optimization is shown in Figure 11.

The starting point for searching for impedance *Z_LT_* is the load impedance *Z_L_* which is a compromise between maximum output power and maximum power-added efficiency (PAE) impedance. The impedance search area is narrowed down to impedances that meet the following conditions:(13)$$P_{out}\geq 0.7P_{outmax}\;PAE\geq 0.8PAE_{max}$$

It is assumed in the optimization that all harmonics are shorted. The values of $Z_{S}$ and $Z_{L}$ source and load impedances seen by transistor before and after optimization are shown in Table 2.

Assembly drawing of amplifier optimized for minimum transmittance changes is shown in Figure 12.

Pulses shapes as visible result of transmittance variations after and before optimization are shown in Figure 13.

Transmittance variations before and after optimization with obtained output power and PAE of the test amplifier are presented in Table 3.

The simplified guideline for the proposed design method is presented in Figure 14.

## 5. Conclusions

The paper presents the method to improve the amplifier transmittance flatness during pulse as well as other time-varying parameters of amplified signals without deterioration of other significant amplifier parameters such as output power, PAE and gain. The method comprises the Envelope simulations and sophisticated measurement. We simulated the transmittance phase and amplitude in time, e.g., during the pulse. The results of such simulations were consistent with the measurement results. The formulas for the amplifier transmittance were derived. These formulas enable the power amplifiers with minimum transmittance phase changes to be easier designed. For the transparency of this work, presented results relate to simple pulses, but the calculation can be done for a more complex radio signal, like 5G. We are currently working on the application of the presented method to this kind of signal. By applying new design approach, it was possible to improve phase changes on test pulses from 0.5° to 0.2° and decrease amplitude variation from 0.8 dB to 0.2 dB during the pulse width of 40µs and 40% duty cycle with the 17 W of output power and PAE more than 62%. Though we used GaN-on-Si HEMT, the results are very promising, and we are currently testing and modeling amplifiers with GaN-on-SiC HEMTs.

## Figures and Tables

**Figure 1 micromachines-11-00398-f001:**
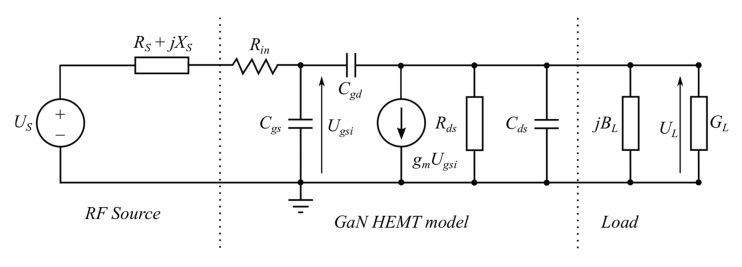
Power amplifier model with GaN high electron mobility transistors (HEMT).

**Figure 2 micromachines-11-00398-f002:**
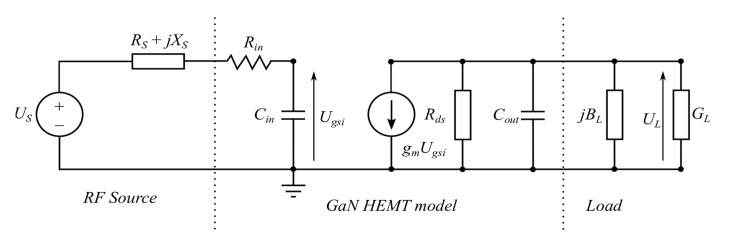
Amplifier model simplified by the Miller theorem.

**Figure 3 micromachines-11-00398-f003:**
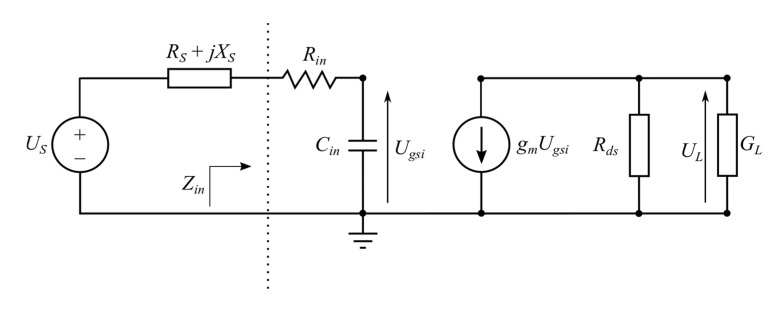
Simplified power amplifier model with GaN HEMT.

**Figure 4 micromachines-11-00398-f004:**
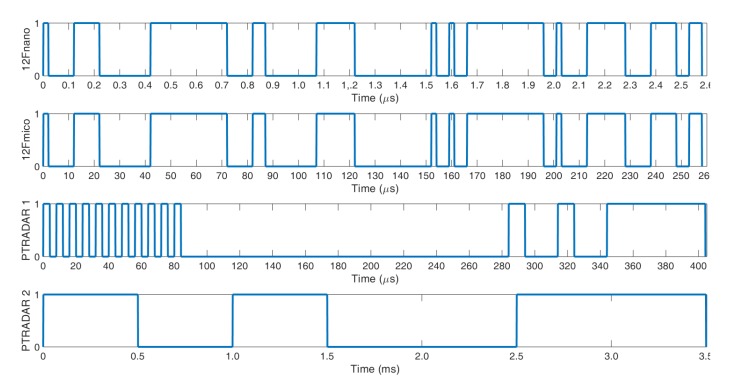
Train pulses used for power amplifier measurements.

**Figure 5 micromachines-11-00398-f005:**
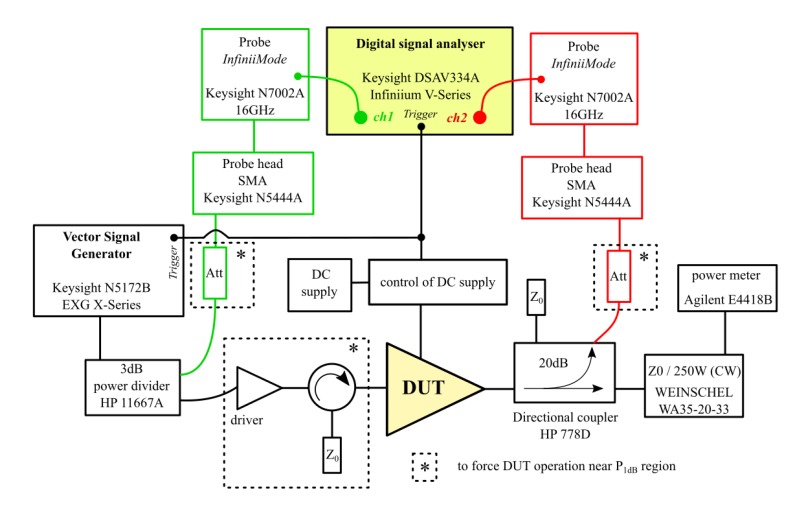
Measurement setup used for transmittance characterization with pulse trains.

**Figure 6 micromachines-11-00398-f006:**
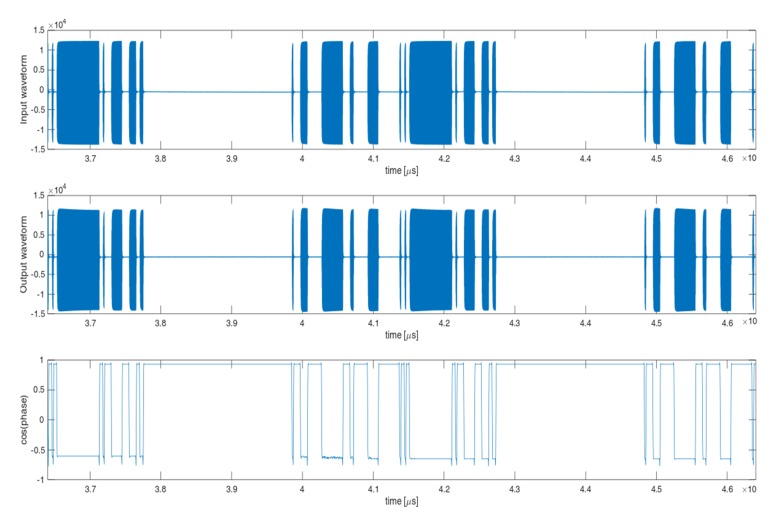
Sample measurement results: input waveform, output waveform and phase difference between waveforms above.

**Figure 7 micromachines-11-00398-f007:**
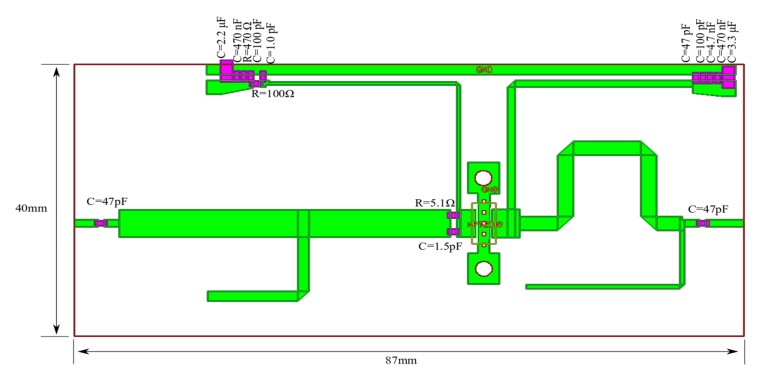
Assembly drawing of the test PA with GaN HEMT NTP2018.

**Figure 8 micromachines-11-00398-f008:**
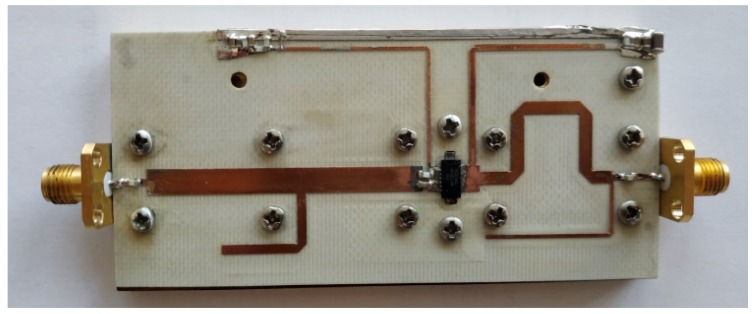
Photography of the test PA with GaN HEMT NPT2018.

**Figure 9 micromachines-11-00398-f009:**
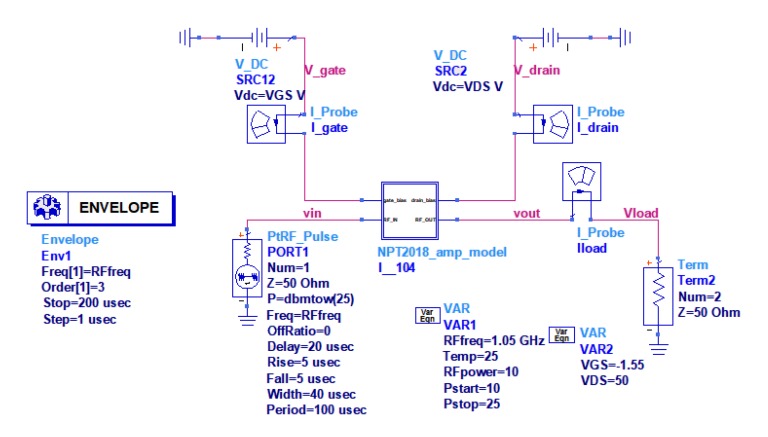
Simulation ADS schematic for modeling of PA transmittance.

**Figure 10 micromachines-11-00398-f010:**
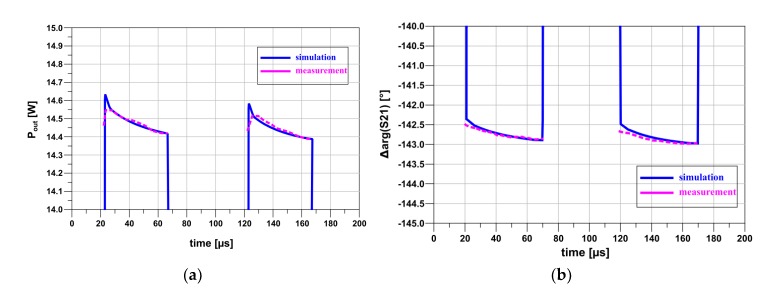
Transmittance changes of the amplifier: (**a**) amplitude changes, (**b**) phase changes during pulse for carrier frequency f = 1.05 GHz.

**Figure 11 micromachines-11-00398-f011:**
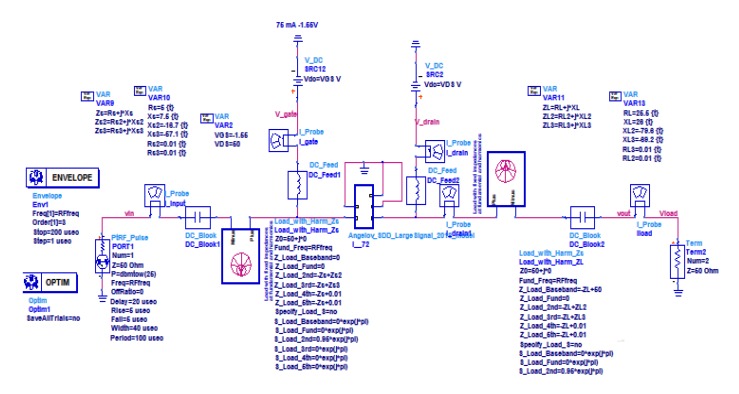
ADS schematic for source and load impedance optimization.

**Figure 12 micromachines-11-00398-f012:**
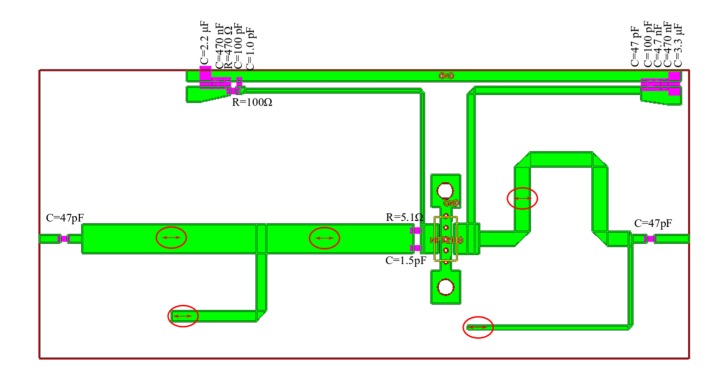
Assembly drawing of the PA with GaN HEMT NTP2018 optimized for minimum transmittance changes, modifications of the matching networks in relation to the previous version of the PA are marked in red.

**Figure 13 micromachines-11-00398-f013:**
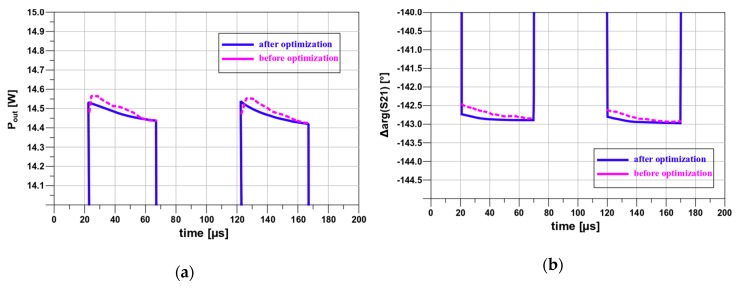
Transmittance changes of the amplifier after and before optimization: (**a**) amplitude changes and (**b**) phase changes during pulse for carrier frequency f = 1.05 GHz.

**Figure 14 micromachines-11-00398-f014:**
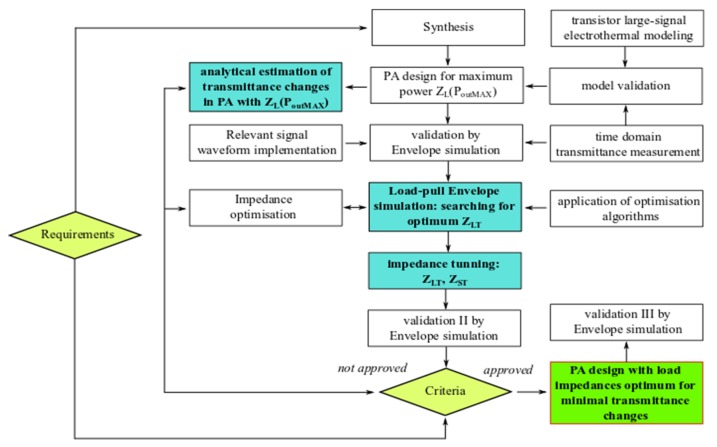
Simplified guidelines for proposed design methodology.

**Table 1 micromachines-11-00398-t001:** Selected parameters for three different GaN HEMTs calculated at the average current for corresponding to *P_sat_*.

Parameter	NPTB0004A	NPT2018	NPT2022
*I_DA_*	0.34 A	0.6 A	4 A
*P_Sat_*	5 W	16 W	120 W
*G′_L_*	16 mS	14 ms	101 ms
*g_m_*	670 mS	584 ms	3.88 S
*G′_L_/g_m_*	0.024	0.025	0.026
*C_gs_*	6.9 pF	7.8 pF	42.9 pF
*C_gd_*	0.27 pF	0.291 pF	1.66 pF
*C_ds_*	0.62 pF	0.56 pF	9.9 pF
*U_DS_*	28 V	50 V	48 V

**Table 2 micromachines-11-00398-t002:** Source and load impedances after and before optimization.

	Original	After Optimization
$Z_{L}\left(f_{0}\right)\;\left[\Omega \right]$	26.4 + 26.6j	31.8 + 35.7j
$Z_{S}\left(f_{0}\right)\;\left[\Omega \right]$	5.6 + 12.0j	5.2 + 17.5j

**Table 3 micromachines-11-00398-t003:** Transmittance changes before and after optimization.

	$\Delta arg\left(S_{21}\right)\left[{}^{\circ}\right]$	$\Delta \left|S_{21}\right|\left[dB\right]$	$P_{out}\left[W\right]$	PAE[%]
Before optimization	0.5	0.8	17 W	61
Optimization applied	0.2	0.2	14 W	54

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
