# Peer review of "A GaN HEMT Amplifier Design for Phased Array Radars and 5G New Radios"

_micromachines, 2020, doi:10.3390/mi11040398_

Round 1
Reviewer 1 Report
It is a paper on circuit design for power amplifier applied to AESA system and 5G radio. I recommend this paper because it is necessary for 5G technology and the results are excellent.
Please correct some spelling, etc. and review the references.
Author Response
Dear Reviewer,
Please find the attached reply.
Kind Regards,
Authors

Reviewer 2 Report
Some points need to be addressed in order to improve the quality of the paper.
GENERAL COMMENTS
It is unclear how section “PA analysis” contributes to the rest of the work. More specifically, how equations 10 and 11 are used to design the PA?
Consequently, the Authors do not sufficiently explain how they designed the “first version” of the PA presented in section 3.
At the beginning of section 4 the Authors mention “… proposed design methodology”. If a design methodology is presented, then it should be clearly illustrated by providing a step-by-step procedure or at least some form of guideline.
SPECIFIC COMMENTS
Introduction. 5G radios and pulsed Radars operate in very different signal conditions. The Authors are claiming the proposed design procedure is applicable to both scenarios?
Introduction. The Doherty technique – probably the most widespread for high PAPR PAs – is missing in the Introduction. Please verify and comment if necessary.
Section 2. The GaN HEMT model in Fig. 1 does not contain any NON-LINEAR component. It appears to be rather simplified with respect to the considered circuit complexity (highly non-linear PA). Moreover, the model is extracted close to saturated output power, but the circuit is operated in high PAPR condition. Please comment.
Section 2. Please comment on how equations 8 to 11 contribute to the rest of the paper.
Section 3. Please provide some information on how the DUT PA was designed.
Section 3 page 7 line 164. The authors mention a large-signal HEMT provided by MACOM. What are the differences with the model in Fig. 1 and how does the NON-LIN MACOM model contribute to this design?
Fig.7. Please provide x – y dimensions.
Author Response

(The authors gave the same response as above.)

Reviewer 3 Report
Dear Authors, find the attached report, please.

Author Response

(The authors gave the same response as above.)

Round 2
Reviewer 2 Report
This Reviewer appreciates the improvements proposed by the Authors, especially the flow chart in fig 14.
Anyhow, some minor points need to be addressed.
Line 134. Please explain how such reduction was obtained (i.e tuning of values of RS, GL, .. ?) and how they are affected by eq 8 to 11.
ZLT is introduced for the first time in line 226 at page 9/29. Please explain how eqs. 8 to 11 are used to “find starting value of ZLT”.
Author Response

(The authors gave the same response as above.)

Reviewer 3 Report
Thanks to the Authors for a revised paper. Please, find several comments attached.

Author Response

(The authors gave the same response as above.)
